# Host–Microbiota Interactions in Liver Inflammation and Cancer

**DOI:** 10.3390/cancers13174342

**Published:** 2021-08-27

**Authors:** Julie Giraud, Maya Saleh

**Affiliations:** 1ImmunoConcEpT, CNRS, UMR 5164, University of Bordeaux, F-33000 Bordeaux, France; jgiraud@immuconcept.org; 2Department of Medicine, McGill University, Montreal, QC H3G 0B1, Canada

**Keywords:** microbiome, gut-liver axis, inflammation, obesity, alcoholic liver disease, metabolism, innate immunity, cirrhosis

## Abstract

**Simple Summary:**

Hepatocellular carcinoma (HCC) is a difficult to treat liver cancer that generally arises in individuals suffering from alcoholic or non-alcoholic fatty liver diseases. Inflammation, tissue injury and fibrosis are important precursors of HCC. In this review, we explore the links between the microbiota, inflammation and carcinogenesis in the context of HCC. We discuss how the gut and liver communicate and how microbial molecules, including structural components and metabolites, elicit inflammation and tumorigenesis in the liver. A better understanding of microbiota-dependent mechanisms of liver cancer development might lead to novel microbial-based therapeutic approaches.

**Abstract:**

Hepatocellular carcinoma (HCC) is a classical inflammation-promoted cancer that occurs in a setting of liver diseases, including nonalcoholic fatty liver disease (NAFLD) or alcoholic liver disease (ALD). These pathologies share key characteristics, notably intestinal dysbiosis, increased intestinal permeability and an imbalance in bile acids, choline, fatty acids and ethanol metabolites. Translocation of microbial- and danger-associated molecular patterns (MAMPs and DAMPs) from the gut to the liver elicits profound chronic inflammation, leading to severe hepatic injury and eventually HCC progression. In this review, we first describe how the gut and the liver communicate and discuss mechanisms by which the intestinal microbiota elicit hepatic inflammation and HCC. We focus on the role of microbial products, e.g., MAMPs, host inflammatory effectors and host–microbiome-derived metabolites in tumor-promoting mechanisms, including cell death and senescence. Last, we explore the potential of harnessing the microbiota to treat liver diseases and HCC.

## 1. Preface

Hepatocellular carcinoma (HCC) is a classical inflammation-associated cancer that occurs in a setting of alcoholic or nonalcoholic fatty liver diseases (ALD and NAFLD, respectively). Alcohol intake and the metabolic syndrome, along with chronic infection by hepatitis B or C viruses (HBV or HCV) or immune dysfunction, such as in primary biliary or sclerosing cholangitis, represent the main risk factors of HCC [1]. Whereas the main feature of non-progressive forms of ALD and NALD is steatosis (i.e., fat accumulation in hepatocytes), progressive forms, namely alcoholic and non-alcoholic steatohepatitis (ASH and NASH, respectively), also exhibit chronic inflammation, liver injury, fibrosis and, in most cases, cirrhosis, which is characterized by excessive fibrous scars and regenerating nodules. These liver diseases share key characteristics, including intestinal dysbiosis; increased intestinal permeability; and an imbalance in bile acids, choline, free fatty acids and ethanol metabolites. Collectively, such alterations, along with the translocation of microorganisms and/or microbial-associated molecular patterns (MAMPs) to the liver, elicit severe inflammation, hepatic injury and fibrosis. In such a setting, compromised liver function results in the accumulation of tumor-promoting metabolites that, together with oncogenic mutations, drive cirrhosis progression to HCC. In this review, we discuss recent studies characterizing the gut–liver axis and focus on the role of the intestinal microbiota in promoting liver diseases.

## 2. Gut–Liver Communications

The liver is intimately linked to the gut and represents a critical metabolic hub involved in digestion, detoxification and clearance of microbial products. Its building blocks are the hepatic lobules organized around central veins and portal triads, consisting of a portal vein, a hepatic artery and a bile duct. The portal vein delivers 80% of the total liver blood supply, while the remaining 20%, i.e., the oxygenated blood, flows through the hepatic artery. Upon mixing, the blood flows across the lobule through the hepatic sinusoids and drains into the central veins, while the bile flows in the opposite direction via the bile canaliculi. Such an organization establishes gradients of oxygen and metabolites and creates a liver zonation. The hepatocytes, endothelial and immune cells of the liver are aligned along this vasculature, and their spatial distribution along the liver zonation dictates their phenotypes and functions. This has been recently confirmed by single-cell analyses [2] and functional studies [3]. For instance, Gola et al. demonstrated that zonation of Kupffer cells (KCs) is controlled by the commensal microbiota that instruct a CXCL9 chemokine gradient controlling not only the spatial distribution of these liver resident macrophages but also their function in host defense [3]. The gut–liver communication axis involves multiple systems and messengers, namely (1) the biliary tract that delivers primary bile acids (BA) and antimicrobial molecules, including immunoglobulin A (IgA) to the intestine; (2) the portal vein that carries MAMPs and secondary BA to the liver; and (3) the systemic circulation that distributes dietary metabolites, free fatty acids, ethanol and choline metabolites to the intestine across blood capillaries (Figure 1). This gut–liver bi-directional connection regulates the intestinal microbiota composition and the integrity of intestinal epithelial and vascular barriers, as well as BA synthesis and composition and lipid metabolism in the liver. 

## 3. The Central Role of the Gut Microbiome in Liver Diseases 

At birth, we are colonized by a collection of microorganisms, including bacteria, fungi, viruses and archaea, that outnumber our human cells by 10:1 and provide 100× as many genes as found in our human genome [4,5]. While several of our tissues are colonized by specific microbiota, the largest concentration is found in the intestine. The gut microbiota provide essential functions for our digestion and nutrient absorption, including the breakdown of indigestible carbohydrates, the synthesis of vitamins and the deconjugation of primary BA, and in shaping our mucosal immune system. Among the multiple environmental factors that converge on microbiota variations [6], ethnicity and geographical localization exert the most dominant effects [7]. Nonetheless, diet is also an important modulator of microbiota composition [8]. Indeed, early studies reported a dysbiotic microbiota, with a low Bacteroidetes/Firmicutes ratio [9] and a reduced microbiome diversity [10,11] in obese individuals consuming a high-fat diet (HFD). These microbiota alterations were associated with intestinal barrier dysfunction, lipopolysaccharide (LPS) endotoxemia and metabolic adaptations [12,13,14]. However, since each phylum contains both beneficial and deleterious microorganisms, the next step was to functionally determine which components contributed to health versus disease. For instance, the mucin-degrading bacterium *Akkermansia muciniphila* of the Verrucomicrobia phylum emerged as a species associated with a healthy state, reduced obesity and the ability to counter the metabolic abnormalities and inflammatory effects of HFD feeding [15]. 

Altered intestinal microbiota, accompanied by small intestinal bacterial overgrowth (SIBO), are observed in liver chronic inflammatory diseases [16,17,18], cirrhosis [19,20,21,22,23] and HCC [24], and gut microbiome metagenomic signatures have been identified in patients with NAFLD [25], cirrhosis [17] and HCC [26]. For instance, Boursier et al. reported that fecal *Bacteroides* and *Ruminococcus* were independently associated with NASH and fibrosis (stage 2 or higher), respectively, while *Prevotella* was depleted in these conditions [21]. Loomba and colleagues showed an increased abundance of *Bacteroides vulgatus* and *Escherichia coli* in NAFLD patients with advanced fibrosis [25]. Of note, *E. coli* is the predominant bacterium detected by culture in NAFLD patients exhibiting SIBO [16]. In mice, prolonged high dietary cholesterol feeding caused spontaneous NAFLD and progression to HCC development, which was associated with gut-microbiota dysbiosis [27]. Importantly, treatment with a cholesterol-lowering drug restored the microbial ecology and prevented disease development in mice [27]. In mouse models of NASH or obesity-induced HCC, *Akkermansia* spp., *Prevotella* spp. and *Lactobacillus* spp. are decreased, while *Bacteroides* spp., *Clostridium* spp. and *Ruminococcus* spp. are elevated [28,29,30] (Figure 2). Enrichment of *Enterobacteriaceae* and depletion of *Lactobacilli* and *A. muciniphila* were also seen in ALD and ASH patients, where the decrease in *A. muciniphila* indirectly correlated with disease severity, and oral supplementation of *A. muciniphila* promoted intestinal barrier integrity and ameliorated ALD in mice [31]. At the genus level, independent studies reported a cirrhosis-associated over-representation of *Veillonella* in the duodenum [22] and colon [17,19,23,24], and a reduction of *Akkermansia*. In HCC, the expansion of *Bacteroides* and *Ruminococcaceae* correlates with calprotectin concentrations and systemic inflammation [18,24,32,33]. Interestingly, the gut-microbiome bacterial signatures alone [19] or in combination with patient age, sex and body mass index [17] or serum albumin level [23] can be viewed as a non-invasive tool that can predict cirrhosis and early HCC, as demonstrated in a large cohort of HBV-infected patients in China [34]. Last, besides bacteria, the intestinal microbiota diversity is reduced in patients with liver diseases, particularly ALD patients, in whom candida dominates [35]. 

The key role of the microbiota in HCC was functionally demonstrated with the use of antibiotics, which reduce liver inflammation and tumor development in mouse models [28,36]. Besides promoting hepatic inflammation, the microbiota could also impact antitumor immunity. Notably, some species, including *A. muciniphila* and *Ruminococcaceae* spp., were found to be enriched in the gut of HCC patients that respond to anti-PD-1 immune-checkpoint blockade compared to non-responders [37]. Interestingly, the microbial metabolite 3-indopropionic acid (IPA) was recently shown to enhance the tumorilytic activity of γδ T cells in a murine model of HCC [38]. However, additional studies are warranted to fully understand how the microbiota control antitumor immunity.

Last, besides the gut microbiota, a local microbiome may also influence liver disease severity and tumor development. Indeed, a diverse repertoire of bacterial DNA is detected in the liver of NAFLD patients, albeit at low amounts (up to 2.5 10^4^ reads counts per 12.5 ng of total liver DNA after filtering strategies). In particular, Gamma-proteobacteria are associated with liver disease severity, progressively increasing from non-NAFLD to NASH [39]. An early study exploring the presence of *Helicobacter* species in the liver of patients with HCC reported 16S rDNA-based detection of these bacteria in 8 out of 20 HCC human tumor tissues [40]. This finding has been supported and expanded by the work of Rob Knight and colleagues, who analyzed microbial reads in whole-transcriptome sequencing datasets from TCGA cohorts, including LIHC [41]. It appears that a largely uncharacterized local tumor microbiome exists in HCC that can potentially modulate tumorigenesis and antitumor immunity, akin to what was shown in pancreatic cancer [42].

## 4. Intestinal Permeability as a Precursor of Liver Diseases 

Early studies have reported a link between liver diseases and disrupted intestinal epithelial cell (IEC) barrier integrity. For instance, NAFLD patients with no significant alcohol consumption were shown to exhibit disrupted IEC tight junctions and increased prevalence of SIBO, which correlated with the severity of steatosis [43]. Similarly, both in vitro and in vivo studies demonstrated the impact of alcohol on enhancing intestinal permeability by altering the expression of the tight junction proteins zonula occludens-1 (ZO-1) and claudin-1 [44,45]. Consistently, occludin deficiency in mice led to a more severe ALD phenotype [45]. Besides the IEC barrier, a gut vascular barrier (GVB), as described by the Rescigno lab [46], also restricts liver injury and the translocation of bacteria or bacterial products into the systemic circulation. The GVB is composed of intestinal endothelial cells closely associated with pericytes and enteric glial cells and is thought to provide an additional protective layer that shields the liver from microbial moieties and inflammatory damage [47]. The integrity of the GVB is sustained by the activation of the WNT/β-catenin pathway, which inhibits the expression of the plasmalemma-vesicle-associated protein 1 (PV-1) [46], a membrane glycoprotein associated with the structure of fenestrated endothelia [48], and upregulated in leaky GVB [46] (Figure 1). It is posited that gut microbiota dysbiosis results in reduced BA-mediated stimulation of the farnesoid X receptor (FXR) that drives β-catenin activation in endothelial cells. This has been demonstrated in mouse models of NASH induced by a HFD or a choline and methionine deficient (MCD) diet [49], and a mouse model of cirrhosis induced by bile duct ligation combined with carbon tetrachloride (CCL4)-mediated liver injury [50]. In both models, intestinal permeability led to microbial translocation from the gut to the liver, and FXR agonists, such as the BA analog obeticholic acid (OCA), conferred protection against GVB disruption, limited bacterial translocation and ameliorated liver pathology [49,50]. GVB disruption, as monitored by elevated PV-1 expression, is also observed in patients with colorectal cancer (CRC), and it appears to be triggered by CRC intra-tumoral bacteria. Upon GVB impairment, bacteria disseminate to the liver and prepare a premetastatic niche that promotes the recruitment of metastatic cells from primary CRC [51]. Such a pre-metastatic niche and distant metastases are prevented with antibiotic treatment in mice, suggesting that bacteria and bacterial products migrate to the liver before tumor cells arrive [51]. 

## 5. MAMPs and PRR Activation Link Microbiota Dysbiosis to Liver Inflammation and HCC

MAMPs reaching the liver, including LPS, unmethylated cytosine–phosphate–guanine dinucleotides (CpG) DNA and lipoteichoic acid (LTA), have long been shown to elicit hepatic inflammation by stimulating Toll-like receptor (TLR)4 [52,53], TLR9 [54] and TLR2 [55,56], respectively. These PRR expressed by liver immune cells, including KCs [53] and hepatic stellate cells (HSCs) [52,54,56], signal through the adaptor MYD88 to activate MAPKs and NF-κB transcriptional programs, leading to oxidative and endoplasmic reticulum stress; cellular senescence; and the secretion of inflammatory cytokines, chemokines, matrix-remodeling factors and growth factors that collectively promote liver tumorigenesis [28,56]. Notably, while TLR4-mediated signaling was demonstrated to promote liver fibrosis and fibrosis-associated tumorigenesis [53,57], TLR2 stimulation by LTA, in conjunction with the action of the secondary BA deoxycholate (DCA; a reported inducer of DNA damage [58] and apoptosis [59], was implicated in obesity-associated HCC [56]. TLR2 signaling in HSCs led to senescence and the establishment of a senescence-associated secretory phenotype (SASP), which promoted obesity-induced HCC through a pro-inflammatory and immunosuppressive pro-tumorigenic environment involving prostaglandin E (PGE)2 overproduction [56]. Consequently, a BAs sequestrant that formed non-absorbable complexes in the intestine reversed liver injury and prevented NASH progression [60], as well as NASH-HCC development in mice [30] (Figure 2).

In contrast to TLRs role in liver diseases, inflammasome signaling counters NASH development by controlling intestinal homeostasis through IL-18 [61] and gut microbial ecology [62,63]. *Nlrp3^−/−^* or *Nlrp6^−/−^* mice fed a MCD diet [64] or a HFD [65] were more susceptible to NASH compared to WT controls. Mechanistically, impaired inflammasome signaling resulted in the accumulation of TLR4 and TLR9 agonists in the portal circulation leading to enhanced hepatic TNF production, liver inflammation and NASH progression [64]. Such an inflammasome-deficient mouse phenotype was shown to be transferable through co-housing experiments, pointing to a key role of the microbiota in mediating disease severity [64]. Last, and consistent with microbiota dysbiosis and candida expansion reported in ALD patients, β-glucan was shown to elicit hepatic inflammation through PRR activation, particularly the stimulation of C-type lectin domain family 7 member A (CLEC7A) signaling in KC [35]. Importantly, antifungal agents ameliorated ALD in an ethanol-induced liver injury mouse model [35]. 

## 6. Host–Microbiome Metabolic Interactions in Liver Pathologies: A Focus on BA, SCFA, Choline and Ethanol Metabolites

*Bile acids.* Primary BAs are synthetized in hepatocytes from cholesterol through the action of Cholesterol 7 alpha-hydroxylase, also known as cytochrome P450 or CYP7A1 [66]. In humans, the main primary BAs are cholic acid (CA) and chenodeoxycholic acid (CDCA), while CDCA is further metabolized into beta-muricholic acid (betaMCA) in mice [67]. Primary BAs are conjugated mainly to glycine or taurine before getting secreted in the gall bladder, along with phospholipids, to form bile salts. These are released in the duodenum to facilitate emulsification and absorption of lipids and fat-soluble vitamins. BAs are de-conjugated by microbial bile salt hydrolases (BSH) before getting reabsorbed in the terminal ileum and recirculated back to the liver. Around 5% of primary BAs escape reabsorption and transit to the colon, where anaerobic bacteria, particularly Gram-positive bacteria belonging to the Clostridium cluster XI and XIVa, convert them through 7alpha-dehydroxylation into secondary BAs, namely DCA and lithocholate (LCA), which are transported back to the liver via the enterohepatic circulation [68]. BAs are considered hormones that regulate host metabolism and contribute to the shaping of host immunity [69]. Their production is tightly regulated by BA receptors, FXR and TGR-5 (Takeda G protein Receptor 5) [70,71]. For instance, through endocrine production of fibroblast growth factor (FGF19 and FGF15 in mice), FXR limits de novo BA synthesis in the liver through suppression of CYP7A1 transcription in hepatocytes [72]. Several studies have reported dysregulated BA metabolism associated with microbiota dysbiosis in liver diseases [73] and HCC [30]. By reducing the expression of BA transporters, liver inflammation was shown to result in intrahepatic retention of BAs that directly promote HCC proliferation [30]. In an ethanol-induced mouse model of ASH, reduced FXR activity and FGF15-mediated inhibition of Cyp7a1 promoted liver injury [74]. Similarly, in a murine model of NASH, antibiotics treatment suppressed HCC and reduced the levels of secondary BAs in the liver, implicating these in tumorigenesis, potentially through mTOR activation in hepatocytes [36]. In addition, secondary BAs were shown to impair antitumor immunity mediated by NKT cells, by decreasing the production of the NKT chemokine CXCL16 by LSECs [75] (Figure 2). Different strategies aimed at restoring BA homeostasis, such as treatment with FXR agonists or FGF19 [74] or BA sequestrants [60], have shown efficacy in improving liver diseases in mouse models. More recently, the role of this biliary network in regulating the Th17-Treg balance in the gut has been described [76,76,77]. 

## 7. Short Chain Fatty Acids (SCFAs) 

SCFAs, which are produced by bacterial fermentation of dietary fibers, have been recently implicated in HCC. SCFAs, including butyrate, propionate and acetate, are generally associated with metabolic health [78]. For instance, in a randomized controlled trial that administered the fermentable fiber inulin coupled to a propionate ester to overweight adults reported decreased abdominal adiposity, lipid accumulation in the liver and body weight gain compared to controls receiving inulin alone [79]. This effect could be mediated by enhanced fat acid oxidation and energy metabolism, as shown in a second trial in which a mixture of SCFAs was infused in the colon of overweight or obese men [80]. Butyrate and propionate production are also altered in ALD. The reduction of butyrate is associated with a weakness of intestinal permeability, while the administration of butyrate in the form of tributyrin reduced intestinal permeability and subsequent liver injury in ethanol-fed mice [81]. However, this notion has been recently challenged by the group of Vijay-Kumar, who showed that large amounts of SCFA, particularly butyrate in a context of dysbiosis, may instead create a tumor-promoting environment. They reported that, while inulin protected mice from obesity, a longer period of inulin feeding, i.e., over 6 months, promoted cholestatic HCC in 40% of TLR5-deficient mice with a pre-existing dysbiotic microbiota [82]. Soluble-fiber-induced HCC was shown to be microbiota-dependent and transmissible to wild-type (WT) mice in cohousing or cross-fostering experiments. More importantly, interventions that deplete butyrate-producing bacteria, e.g., with metronidazole, inhibit gut fermentation, e.g., via supplementation of plant-derived β-acids, exclude soluble fiber from the diet, or prevent enterohepatic recycling of BAs with cholestyramine reverted inulin-induced HCC in these mice [82]. 

## 8. Choline

Another host–microbiota-controlled metabolite is the macronutrient choline. Choline processing into phosphatidylcholine by the host prevents steatosis. Indeed, feeding mice a choline-deficient diet is a classical model of NASH. Alternatively, choline can be converted into trimethylamine (TMA) by intestinal bacteria, and it is then further metabolized in the liver into trimethylamine-N-oxide (TMAO). In contrast to phosphatidylcholine, TMAO is at the origin of hepatic steatosis by promoting triglyceride accumulation (Figure 2). Increased systemic circulation of TMAO is associated with a reduced level of phosphatidylcholine, and this imbalance is characteristic of patients with NAFLD [83] and NASH [84,85].

## 9. Ethanol

Ethanol metabolism greatly impacts HCC development through different mechanisms. The commensal microorganisms express genes that can ferment dietary sugars into ethanol that is absorbed by the gastrointestinal tract by simple diffusion. Moreover, liver cells, enterocytes and gut microbiota components express alcohol dehydrogenase, which co-metabolizes ethanol into toxic acetaldehyde that is then converted by CYP2E1—if this pathway is not saturated—to acetate. Acetaldehyde has been involved in weakening tight junctions of the intestinal epithelial barrier [45], eliciting gut barrier permeability and enabling translocation of microbial products, as shown in ethanol-fed mice [86]. In addition, ethanol exacerbates oxidative stress and hepatic inflammation and is a known carcinogen [87]. 

## 10. Carcinogenic Compounds Produced by Bacteria in Gastrointestinal Cancers 

Besides the pro-inflammatory effects of the microbiota indirectly promoting cancer, and microbiota-derived pro-tumorigenic metabolites, several reports have demonstrated that some commensal bacteria produce carcinogenic compounds that directly contribute to cancer initiation. Most notable is the causative role of nitrosating bacteria that produce carcinogenic N-nitroso compounds (NOCs) in tumorigenesis [88]. Indeed, various bacteria implicated in HCC, including *E. coli* and *Clostridium* spp., can produce NOCs, and dietary NOCs, including N-nitrosodiethylamine (NDEA) and N-nitrosodimethylamine (NDMA), and increase the risk of developing HCC in hepatitis virus positive patients [89]. The intestinal microbiome is also involved in mycotoxicosis that is associated with HCC [90]. Indeed, mycotoxins, including aflatoxins, which are secondary metabolites produced by fungi, lead to microbiome dysbiosis and intestinal permeability that indirectly promotes liver cancer, as previously discussed. Additional examples of carcinogenic compounds produced by bacteria of the gut microbiome include (a) colibactin produced by the polyketide synthase (*pks*) island of certain *E. coli* species [91]; (b) a zinc-dependent metalloprotease toxin produced by enterotoxigenic *Bacteroides fragilis*, termed *B. fragilis* toxin (BTF) [92]; and (c) a carcinogenic *Fusobacterium nucleatum* adhesion complex encoded by *FadA* [93]. While these bacterial products have been demonstrated to contribute to colorectal cancer, by inducing DNA damage, stimulating STAT3 or β-catenin activation, respectively, their role in liver cancer is less well established. Nonetheless, these examples illustrate additional mechanisms by which the microbiota could drive gastrointestinal tumorigenesis. 

## 11. Conclusions and Perspectives 

There is no doubt that our microbiota are essential for our well-being. Indeed, we are meta-organisms composed of human and microbial cells and meta-genomes [4,5] that act in synergy to regulate our physiology both in the intestine and systemically [94]. However, accumulating evidence now clearly implicates our commensal organisms in diverse pathologies. In this review, we focused on liver inflammatory diseases linked to HCC and (a) discussed evidence demonstrating microbiota dysbiosis preceding or associated with these conditions; (b) discussed the impact of environmental factors, in particular diet, i.e., high-fat, high-fiber or choline-deficient diet, and alcohol consumption in such microbiota alterations; and (c) reported mechanisms linking microbial triggers, be it structural components or metabolites, to hepatic inflammation, tumorigenesis and antitumor immunity. While much progress has been made in this field, for instance, in cataloguing enriched or depleted microorganisms in health versus disease, we now need to move to next-generation studies of the functional microbiome in order to be able to design microbial strategies to counter liver diseases and HCC. The most actionable variable to be precisely evaluated is diet, which could provide therapies based on lifestyle changes, supplements or prebiotics. Another variable to consider is the medications that could converge on the microbiome, e.g., antibiotics and proton pump inhibitors to name a couple. Antibiotics, for example, were demonstrated to blunt patients’ responses to cancer therapies, including immunotherapies [95]. Additional approaches might include the development of microbiome-based biotherapies [96,97] for patients, as an alternative or supplement to fecal microbiota transplantation (FMT), as it has shown efficacy in some conditions, particularly in patients infected with *Clostridium difficile* [98]. The choice of approach can be guided by the recent classification of the human microbial ecology by Hildebrand and colleagues into five categories based on strain persistence and evolution, namely tenacious, spatio-persistent, heredi-persistent, average-persistent and non-persistent strains [99]. Interestingly, the authors propose that, while the first two can be targeted by FMT, chronic reinfection strategies, e.g., with single-strain biotherapies, might be needed to restore heredi-persistent strains [99]. The question is then, can we harness the microbiome to treat liver diseases and HCC? The answer is yes and hopefully soon.

## Figures and Tables

**Figure 1 cancers-13-04342-f001:**
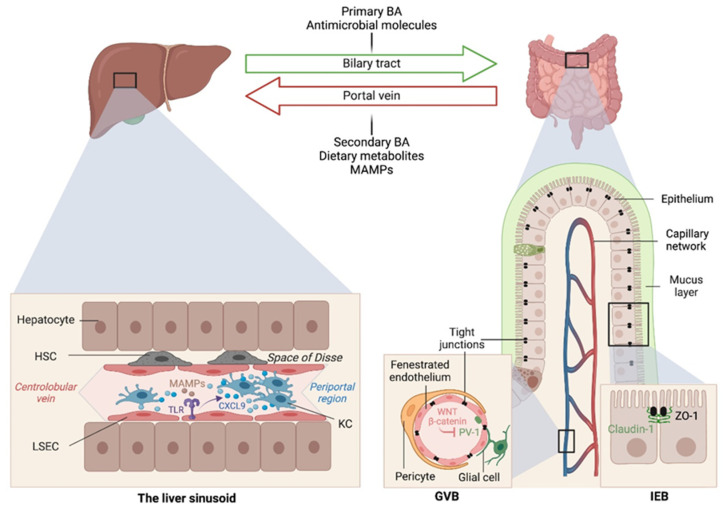
Gut–liver communication axes in health. The gut and liver are intricately connected through the portal vein and biliary circulations. Several messengers, including dietary metabolites, bile acids, antimicrobial peptides and microbial associated molecular patterns (MAMPs), are exchanged and required for intestinal and hepatic homeostasis, as well as host physiology in general. The intestinal microbiome also instructs liver zonation. Microbial sensing by toll-like receptors (TLR) on liver sinusoidal endothelial cells (LSECs) results in a CXCL9 chemokine gradient that attracts Kupffer cells closer to the periportal region to ensure immune surveillance and host defense from invading pathogens arriving through the portal vein circulation. GVB, gut-vascular barrier; IEB, intestinal epithelial barrier; ZO-1, zonula occludens-1; KC, Kupffer cell; HSC, hepatic stellate cell.

**Figure 2 cancers-13-04342-f002:**
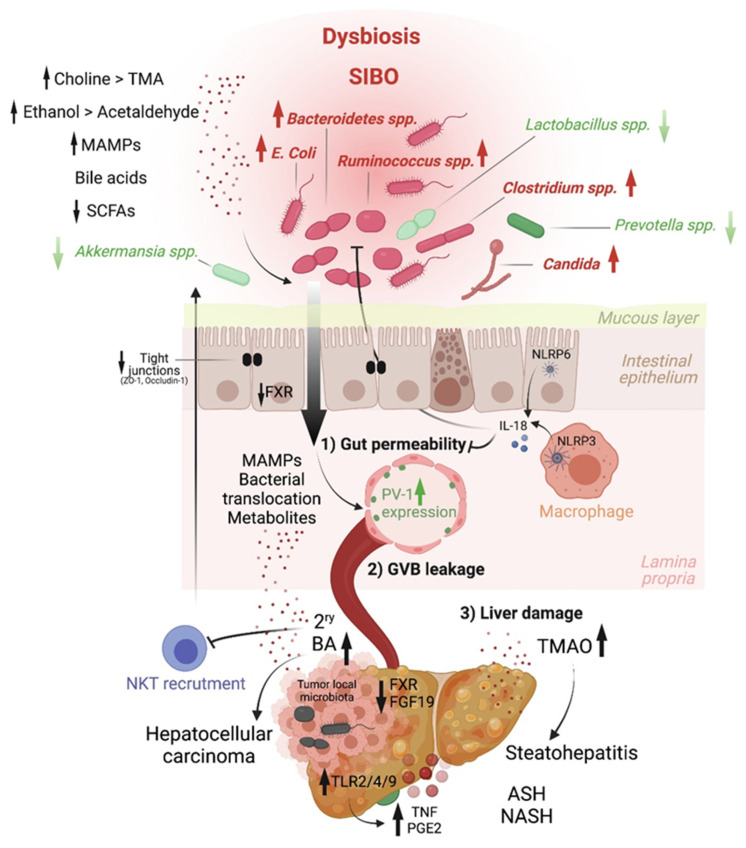
Host–microbiome molecular effectors driving liver inflammation and tumorigenesis. Chronic inflammatory diseases are well-established precursors of hepatocellular carcinoma. Intestinal microbial dysbiosis is associated with these conditions, in which beneficial bacteria (green) are depleted and pathobionts (red) expand. This results in disrupted gut–liver axis homeostasis. On the one hand, a reduction in short-chain fatty acids (SCFA) production leads to increased IEC layer permeability, a process normally countered by the activity of the NLRP3 and NLRP6 inflammasomes via interleukin-18 (IL-18). The integrity of the intestinal epithelial layer can also be compromised by ethanol metabolites. On the other hand, altered bile acids synthesis results in deregulated farnesoid X receptor (FXR)-mediated inhibition of gut-vascular barrier (GVB) leakage (induced by plasmalemma vesicle-associated protein 1 [PV-1]) and secondary bile acids retention in the liver. Last, enhanced choline metabolism to trimethylamine-N-oxide (TMAO) in the liver drives steatohepatitis. The leakage of the intestinal epithelial and gut-vascular barriers allows the translocation of microbial-associated molecular patterns (MAMPs) and microbial components into the liver, triggering Toll-like receptor (TLR)-mediated inflammation (e.g., via tumor necrosis factor (TNF)) and a senescence-associated secretory phenotype (in which prostaglandin E2 (PGE2) is notable), well-known promoters of liver cancer.

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
