# Peer review of "Host–Microbiota Interactions in Liver Inflammation and Cancer"

_cancers, 2021, doi:10.3390/cancers13174342_

Round 1
Reviewer 1 Report
- The concept and correlation in the current review are good.
- The authors have impressively conceptualized the illustrations
- I believe there is a necessity that the authors provide a brief introduction about the liver inflammation before intestinal permeability section.
- The overall article is a well written comprehensive review linking microbiota and HCC via inflammatory link.
Author Response
We thank the reviewer for taking the time to evaluate our review article. We elected not to add a separate paragraph on liver inflammation as the information is provided throughout the text and we wished to maintain the focus on the role of the microbiome.
Reviewer 2 Report
Article is very interesting and good written. I suggest add two thinks:
- Add more information about carcinogenic compounds produced by bacteria in liver cancer.
- In Figure 2, Authors presented microbiota, which has impact on SIBO. Please, add figure with microbiota, which has impact or is present in liver cancer.
Author Response
Thank you for your suggestions, we have now added a paragraph on carcinogenic compounds produced by bacteria and have amended Figure 2 by moving the word SIBO under E. coli. The other microbiome changes shown in the figure illustrate the dybiosis seen in liver diseases (not only SIBO).
Round 2
Reviewer 2 Report
Authors significantly corrected manuscript according to reviewer's suggestions. Recently, I recommend article for publication.